# Effects of pH on Nanofibrillation of TEMPO-Oxidized Paper Mulberry Bast Fibers

**DOI:** 10.3390/polym11030414

**Published:** 2019-03-04

**Authors:** Jung Yoon Park, Chan-Woo Park, Song-Yi Han, Gu-Joong Kwon, Nam-Hun Kim, Seung-Hwan Lee

**Affiliations:** 1The Institute of Forest Science, Kangwon National University, Chuncheon 24341, Korea; cypjy88@gmail.com (J.Y.P.); gjkwon@kangwon.ac.kr (G.-J.K.); 2Department of Forest Biomaterials and Engineering, Kangwon National University, Chuncheon 24341, Korea; chanwoo8973@kangwon.ac.kr (C.-W.P.); songyi618@kangwon.ac.kr (S.-Y.H.); kimnh@kangwon.ac.kr (N.-H.K.)

**Keywords:** paper mulberry bast fiber, cellulose nanofibril, ultrasonication, TEMPO-mediated oxidation, pH, zeta potential, fiber entanglement

## Abstract

TEMPO oxidation was conducted as a pretreatment to achieve efficient nanofibrillation of long paper mulberry bast fibers (PMBFs). The pH dependency of nanofibrillation efficiency and the characteristics of the resulting cellulose nanofibrils (CNFs) were investigated. As the pH increased, the negative value of the zeta potential of TEMPO-oxidized fibers increased. The increase in electrostatic repulsion at pH values of greater than 9 prevented the entanglement of long PMBFs, which was a drawback for defibrillation at acidic pH. With increasing pH, the CNF production yield was increased. The crystallinity index of TEMPO-oxidized CNFs from PMBFs was 83.5%, which was higher than that of TEMPO-oxidized CNFs from softwood fibers in the same conditions. The tensile strength of nanopaper from TEMPO-oxidized PMBF CNFs was 110.18 MPa, which was approximately 30% higher than that (84.19 MPa) of the TEMPO-oxidized CNFs from softwood fibers.

## 1. Introduction

Cellulose nanofibrils (CNFs) are eco-friendly natural polymers with biodegradability and excellent strength, and their applications in various fields have been reported [1,2,3,4,5]. They are generally less than 100 nm in diameter and several micrometers in length and can be produced through various mechanical defibrillation processes, such as high-pressure homogenization [6,7], microfluidization [8,9], grinding [10,11,12], and ultrasonication [13,14,15]. In particular, ultrasonication is known to be adequate for the production of relatively long CNFs [15]. Ultrasonic waves create microbubbles within a liquid and the cavitation effect induces the nanofibrillation of cellulose [16]. Cavitation is the phenomenon of the sequential formation, growth, and collapse of millions of microbubbles in a liquid. During collapse, cavities create high localized temperature and pressure of approximately 5000–10,000 K and 1000–2000 atm, respectively. Chen et al. [15] successfully prepared ultralong and highly uniform CNFs from bamboo fibers with lengths of greater than 1 mm and diameters of 30–80 nm by combining chemical pretreatment with high-intensity ultrasonication. Compared to CNFs of regular length, longer CNFs have advantages such as a high aspect ratio, large specific surface area, and web-like entangled structures for the reinforcement of composite materials [15]. 

Various chemical pretreatments to reduce the energy requirements of mechanical defibrillation have been developed [17,18]. Among them, TEMPO (2,2,6,6-tetramethylpiperidin-1-oxyl radical) oxidation is widely known for the production of homogeneous CNFs with diameters of 3–4 nm [19,20,21,22]. TEMPO oxidation of celluloses regioselectively converts C6 primary hydroxyl groups into carboxylate groups in aqueous media. The strong electrostatic repulsion of cellulose fibers originates from the carboxylate groups introduced by TEMPO oxidation, resulting in high dispersibility of the cellulose suspension and facilitating mechanical defibrillation. More importantly, CNFs prepared by TEMPO oxidation show high transparency and viscosity. 

Paper mulberry bast fibers (PMBFs) have been used in traditional Korean handmade paper (*hanji*) through pulping of the inner bark of the bast. Hanji exhibits excellent strength and preservability because of the long PMBF length of over 8 mm and the high degree of polymerization [23,24,25]. However, fibers that are longer than 3 mm are often entangled during high-shear defibrillation processes [25,26], and such entanglement can occur sequentially, making nanofibrillation difficult in CNF production. Thus, a pre-grinding process to shorten the long fibers has been applied prior to the production of CNFs [27,28,29,30,31], particularly in the case of long non-wooden fibers. 

In this study, TEMPO oxidation was conducted to improve the nanofibrillation efficiency of PMBFs with long fiber length using ultrasonication without pre-grinding to shorten the length, and the effect of pH on the nanofibrillation efficiency of TEMPO-oxidized PMBFs was investigated.

## 2. Materials and Methods

### 2.1. Materials

As a raw fiber, PMBFs were obtained from Andong Hanji (Andong-si, Gyeongsangbuk-do, Korea) and softwood fibers were obtained in the sheet form from Hankuk Paper (Gangnam-gu, Seoul, Korea). Before use, the PMBFs were treated by chemical pulping and the softwood fibers were disintegrated using a lab disintegrator. Potassium hydroxide, hydrogen peroxide, and acetic acid were purchased from Daejung Chemicals & Metals Co. Ltd. (Siheung-si, Gyeonggi-do, Korea). TEMPO (99%, Sigma-Aldrich, St. Louis, MO, USA), sodium bromide (NaBr, 99%, Junsei Chemical Co. Ltd., Tokyo, Japan), and sodium hypochlorite solution (NaClO, 12%, Yakuri Pure Chemicals Co. Ltd., Kyoto, Japan) were used to conduct TEMPO oxidation. Other chemicals were obtained from commercial sources and used without further purification.

### 2.2. Methods

#### 2.2.1. Preparation of PMBFs

For the digestion of paper mulberry bast, dried paper mulberry bast was cut to a length of 50 mm. To prepare the digestion solution, 5.88 g of potassium hydroxide (85%) was dissolved in 200 mL of distilled water. Thereafter, 20 g of paper mulberry bast and 200 mL of digestion solution were placed in a 1-L Erlenmeyer flask at a solid/liquid ratio of 1/10. Digestion was conducted in an autoclave at 120 °C for 4 h. The digested PMBFs were washed with distilled water to remove the digestion solution. Subsequently, disintegration was conducted in a lab disintegrator (Pulp Disintegrator, L&W, Stockholm, Sweden) for 5000 revolutions. The disintegrated PMBFs were washed again through a 200-mesh wire screen or cloth to remove the residual digestion solution and freeze-dried for subsequent experiments.

Extractives were removed following the procedure described in TAPPI T 204. A mixture of ethanol and benzene (1:2 *v*/*v*) was prepared and placed in a 500-mL round-bottom flask. Thereafter, extraction was performed at 90 °C for 9 h in a Soxhlet extractor. For delignification, peracetic acid was prepared by mixing hydrogen peroxide and acetic acid at a ratio of 1:1. Then, 10 g of PMBFs from which extractives were removed and 300 mL of peracetic acid were placed in a 1-L Erlenmeyer flask. The delignification reaction was conducted at 85 °C for 6 h.

#### 2.2.2. TEMPO Oxidation

Cellulose samples (5 g) were suspended in water (1 L) containing TEMPO (0.1 mmol/g cellulose) and sodium bromide (1 mmol/g cellulose). TEMPO oxidation was initiated by adding NaClO solution (5 mmol/g cellulose) and adjusting the pH to 9.8 using 0.1 M HCl. The reaction was performed for 3 h in a pH titrator (TitroLine Easy, SI Analytics, Mainz, Germany), which was maintained at pH 10 using 0.5 M NaOH. To terminate the reaction, ethanol was added to the cellulose suspension and the pH was adjusted to 8.5 with 0.1 M HCl. The TEMPO-oxidized cellulose fibers were washed with distilled water before analysis of fiber property and ultrasonication.

#### 2.2.3. Analysis of Fiber Property

To measure the length and width of the TEMPO-oxidized PMBFs and softwood fibers, a fiber analyzer (Fiber Tester Plus, L&W, Stockholm, Sweden) was used following the standard method of ISO 16065-2. As the device is designed for the analysis of wood fibers, the maximum measurable length is 7.5 mm. In addition, as entanglement is a specific property of long PMBFs, the entangled fibers formed a plug before passing through the imaging window. As a result, it was difficult to properly measure the dimensions of a sufficient number of fibers through the fiber analyzer. Consequently, optical microscopy was used to measure the length and width of TEMPO-oxidized PMBFs. The measurement was conducted at least 500 times.

To measure the change in zeta potential of the TEMPO-oxidized fibers at various pH values, a fiber potential analyzer (FPA, Emtec Electronic, Leipzig, Germany) was employed. The method involved estimating the vacuum pressure, streaming current potential, and electrical conductivity of a fiber plug formed by a vacuum. The zeta potential was determined by the Helmholtz-Smoluchowski equation. Changes in the zeta potential of the fibers were estimated for pH ranging from 2.5 to 11.

#### 2.2.4. Ultrasonication and Yield Analysis

For the nanofibrillation of TEMPO-oxidized fibers, an ultrasound generator (SONIC Dismembrator F550, Fisher Scientific, Hampton, NH, USA) was used to conduct ultrasonication at 20 kHz and 550 W. To do this, 80 mL of a 0.1% cellulose suspension was placed in a 100-mL beaker on ice. Then, the ultrasound probe was placed at a position approximately 1.5 cm away from the surface of the cellulose suspension. After ultrasonication was performed for 30 min, the cellulose suspension was centrifuged at 4000× *g* for 15 min. The yield of cellulose nanofibrils was calculated by the ratio of the dry weights of the supernatant and precipitate. 

#### 2.2.5. Analysis of TEMPO-Oxidized Cellulose Nanofibrils

To analyze the morphological characteristics of the produced TEMPO-oxidized cellulose nanofibrils, the diluted cellulose nanofibrils were dropped onto a 300-mesh carbon grid and stained with uranium acetate. The morphology of the TEMPO-oxidized cellulose nanofibrils was observed by field-emission transmission electron microscopy (TEM, JEM-2100F, JEOL, Tokyo, Japan). From the TEM images, the width of each cellulose nanofibril was measured using ImageJ software (Version 1.52a, National Institutes of Health, Bethesda, MD, USA). 

To examine the crystallinity of the TEMPO-oxidized cellulose nanofibrils, X-ray diffractometer (D/max 2100, Rigaku, Tokyo, Japan) was carried out for reflection measurement using CuKα radiation with conditions of 2θ = 10°–35° and scan speed = 1°/min. The crystallinity index was calculated from Equation (1) based on the Segal method [32]. 

(1)$$\mathrm{Crystallinity}\;\mathrm{Index}\left(\%\right)=\frac{I_{200}-I_{am}}{I_{200}}\times 100$$

Here, *I*_200_ indicates the maximum intensity of the diffraction peak corresponding to 200 planes at 2θ = 22.6° and *I_am_* represents the diffraction intensity at 2θ = 18°.

To evaluate the strength of the sheets formed by TEMPO-oxidized cellulose nanofibrils, a polytetrafluoroethylene membrane filter (ADVANTEC^®^, Toyo Roshi Kaisha Ltd., Tokyo, Japan) with a pore size of 0.45 µm was used for vacuum filtration. After vacuum filtration, the wet-web was dried using a hot press at 105°C and 1 MPa. Dried nanopaper sheets were conditioned at 23 ± 1 °C and 50 ± 2% RH, according to ISO 187, for over 24 h. Then, nanopaper sheets were cut to be 5 mm in width and 40 mm length and then weighed. The thickness was measured by using a digimatic indicator (ID-C112XB, Mitutoyo Co., Kawasaki, Japan). The apparent density was calculated by dividing the weight by the volume. The tensile strength of the dried sheets was measured 7 times for each sample with a universal testing machine (WL2100, WITHLAB Co., Ltd., Gunpo-si, Gyeonggi-do, Korea) at a cross-head speed of 20 mm/min. The samples were 5 mm in width and 20 mm in span length.

## 3. Results and Discussion

### 3.1. Fundamental Characteristics of TEMPO-Oxidized Fibers

Figure 1 shows the fiber length distribution of TEMPO-oxidized PMBFs and softwood fibers. The distribution of the length TEMPO-oxidized PMBFs was broader than that of TEMPO-oxidized softwood fibers. The average length and diameter were 10.81 mm and 22.3 µm for TEMPO-oxidized PMBFs and 2.34 mm and 32.7 µm for softwood fibers, respectively. PMBFs are generally known to be longer than 8 mm, which is considerably longer than wood fibers [25]. Longer fibers can easily cause entanglement during the defibrillation process, even with weak shear force, making it difficult to achieve homogeneous nanofibrillation.

Figure 2 shows the changes in the zeta potential of TEMPO-oxidized PMBFs and softwood fibers at different pH values. The negative value of the zeta potential was dramatically increased at pH 2.5 to 5, whereas its value was decreased at pH 5 to 7. At pH 7, the zeta potential gradually increased until pH 10, and dramatically increased at pH 11. The changes in the zeta potential at different pH values showed similar trends in both PMBFs and softwood fibers. However, the negative values of zeta potential were higher in TEMPO-oxidized PMBFs than those in softwood fibers. In general, cellulose fibers are anionic in water because of the acetyl group from hemicellulose or the functional groups introduced during pulping and bleaching [33]. Notably, the carboxyl group introduced by TEMPO oxidation may be converted into carboxylic acid or a carboxylate ion based on the change in pH, substantially influencing the surface charge of cellulose fibers and the fiber dispersibility in water. At pH 2.5, the carboxyl group introduced by TEMPO oxidation was considered to have converted into carboxylic acid, thus decreasing the number of negatively charged functional groups on the surface. On the other hand, the zeta potential values at pH 3 to 5 were unexpectedly lower than those at alkaline conditions. It is suggested that the carboxylate ion and chloride ions simultaneously affected the high negative zeta potential. Carboxylic acid and carboxylate ions are present at a ratio of approximately 50:50 in the pH range of pH 4–5. In addition, chloride ions adhered to fiber pads during the measurement of zeta potential, thereby contributing to the strong negative charge [34]. In contrast, the carboxyl group exists in the form of carboxylate ions in alkaline conditions, thus increasing the number of negatively charged functional groups and resulting in the increase in zeta potential. In particular, the abundance of carboxylate ions drastically increased the zeta potential at pH 11, which was beneficial in improving defibrillation efficiency. 

### 3.2. Effect of pH on Nanofibrillation

Figure 3 displays the TEMPO-oxidized PMBF suspensions after ultrasonication at different pH values. Since a higher zeta potential appeared at pH 3 and 5, efficient dispersibility of TEMPO-oxidized PMBFs was expected. However, the entanglement of TEMPO-oxidized PMBFs was prominent at pH 3 and 5. It is considered that the low zeta potential at acidic conditions did not originate from the carboxylate ions and this may reduce electrostatic repulsion [22,35]. In contrast, entanglement was significantly decreased at pH values of greater than 7, even though small amounts of entangled fibers were observed at pH 7 and 9. Encouragingly, defibrillation was very effective at pH 11, without showing entangled fibers. At above pH 7, the increased zeta potential due to the carboxylate ions improved fiber dispersibility and decreased the number of entangled fibers. Compared to the long PMBFs, the short TEMPO-oxidized softwood fibers did not show any entanglement at all pH values.

Figure 4 shows an optical microscopic images of entangled TEMPO-oxidized PMBFs after ultrasonication at pH 3. Even with prolonged ultrasonication, it was difficult to defibrillate the fibers once they became entangled. This entanglement prevented uniform exposure of fibers to ultrasound and thus, the nanofibrillation efficiency was reduced. 

Figure 5 shows the weight ratios of the supernatant to the precipitate after the ultrasonicated TEMPO-oxidized fiber suspensions were centrifuged. The supernatant contained the fully individualized TEMPO-oxidized CNFs after ultrasonication, whereas the precipitate contained the entangled or inefficiently fibrillated fibers. With increasing pH, the supernatant proportion of TEMPO-oxidized PMBFs was increased, to 88% at pH 11. In the case of TEMPO-oxidized softwood fibers, the supernatant proportion was approximately 85% at pH 7 and increased to 90% at pH 11. This result indicates that the longer PMBFs were more difficult to defibrillate than the shorter softwood fibers. In addition, the higher zeta potential caused by the generation of carboxylate ions at alkaline conditions may not only prevent fiber entanglement due to long fiber length, but also improve the nanofibrillation efficiency.

### 3.3. Characteristics of TEMPO-Oxidized CNFs

Figure 6 shows representative TEM images of the TEMPO-oxidized CNFs from PMBFs and softwood fibers. The average diameter of the CNFs from PMBFs and softwood fibers was 5.46 and 6.65 nm, respectively. Cao et al. (2012) [36] reported that the diameter of cellulose nanocrystals from TEMPO-oxidized jute fibers ranged from 3 to 10 nm. Puangsin et al. (2013) [37] reported that the average diameters of TEMPO-oxidized CNFs from hemp bast, bamboo, and bagasse pulp were 2.9, 2.4, and 2.5 nm, respectively. Compared with both results using a high-pressure homogenizer for defibrillation, the CNFs in this study were slightly larger.

Figure 7 illustrates the X-ray diffractograms of TEMPO-oxidized CNFs from PMBFs and softwood fibers. The crystallinity index of TEMPO-oxidized CNFs from PMBFs and softwood fibers was 83.5% and 79.2%, respectively, without showing the decreasing effect of cellulose crystallinity by ultrasonication. The difference between the two types of CNFs may be attributed to the higher crystallinity of the original cellulose in PMBFs than that in softwood fibers [23,38].

### 3.4. Tensile Properties

The tensile strength of nanopaper sheets from TEMPO-oxidized CNFs from TEMPO-oxidized PMBFs and softwood fibers was 110.18 and 84.19 MPa, respectively (Table 1). In this study, nanopaper sheets were formed in a relatively mild temperature (105 °C) and pressure (1 MPa) conditions. As the result, apparent density of sheets showed about 1.29 g/cm^3^. Puangsin et al. (2013) [37] reported the tensile strength of 137–232 MPa of the nanopaper sheet from TEMPO-oxidized cellulose nanofibrils with high apparent density (1.41–1.65 g/cm^3^). Fukuzumi et al. (2013) [39] also reported the 224–266 MPa of tensile strength of the nanopaper sheet with 1.40–1.43 g/cm^3^ of the apparent density. In particular, Tarrés et al. (2017) [40] reported a broad range of tensile strength (62.6–152.6 MPa) depending on the apparent density (1.19–1.43 g/m^3^) of nanopaper sheets. According to these studies, the apparent density of sheets is closely related to the tensile strength of nanopaper sheets. Therefore, the low tensile strength in this study may be due to the low apparent density of nanopaper sheets.

## 4. Conclusions

The pH dependency of the nanofibrillation efficiency of TEMPO-oxidized PMFBs with long fiber length was investigated. Entanglement of long fibers occurred during defibrillation at acidic conditions, whereas the nanofibrillation efficiency was improved at alkaline conditions (pH 9–11) because of the increase in electrostatic repulsion, as indicated by the larger zeta potential. The cellulose crystallinity and tensile strength of the TEMPO-oxidized CNFs from PMFBs were higher than those from softwood fibers. The results of this study suggest an effective route to nanofibrillate long non-wooden fibers without fiber entanglement. 

## Figures and Tables

**Figure 1 polymers-11-00414-f001:**
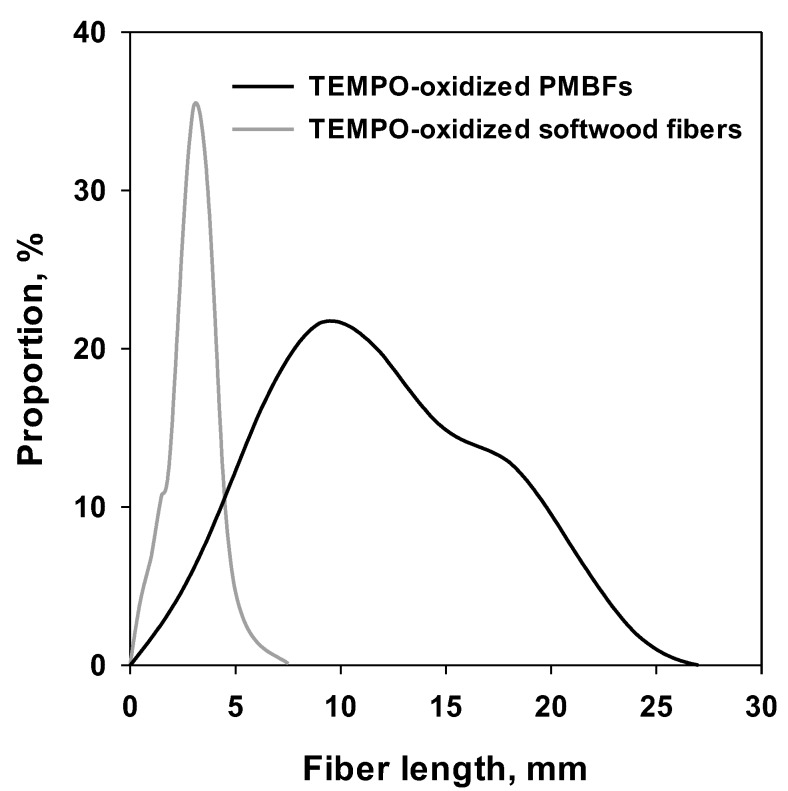
Fiber length distribution of TEMPO-oxidized PMBFs and softwood fibers.

**Figure 2 polymers-11-00414-f002:**
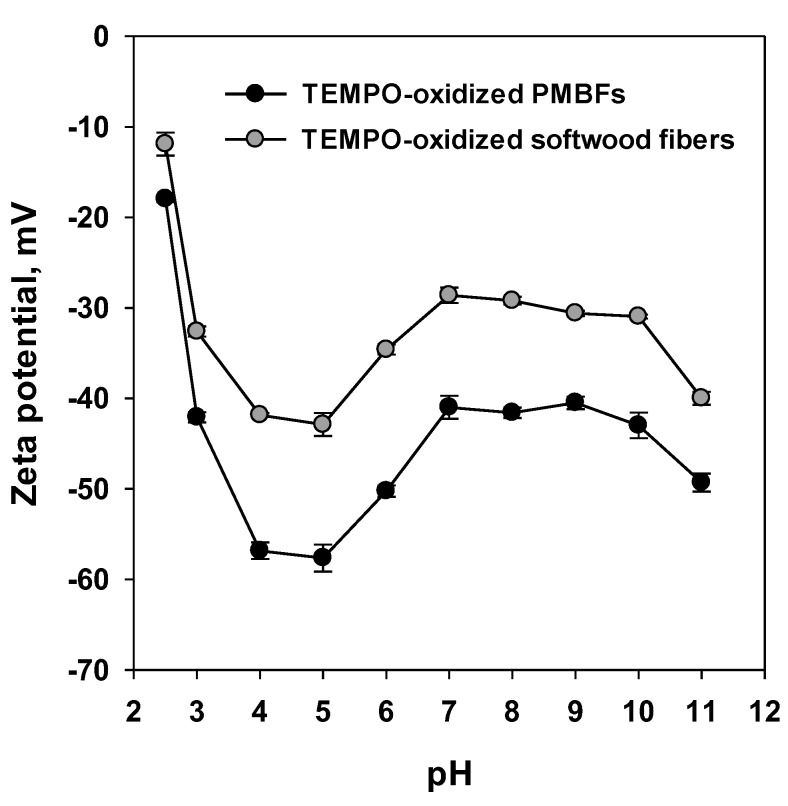
Zeta potential of TEMPO-oxidized PMBFs and softwood fibers at different pH.

**Figure 3 polymers-11-00414-f003:**
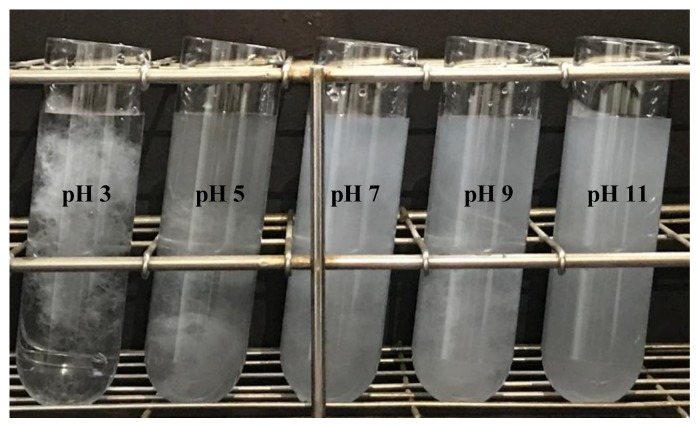
Images of TEMPO-oxidized PMBF suspensions after ultrasonic treatment at different pH values.

**Figure 4 polymers-11-00414-f004:**
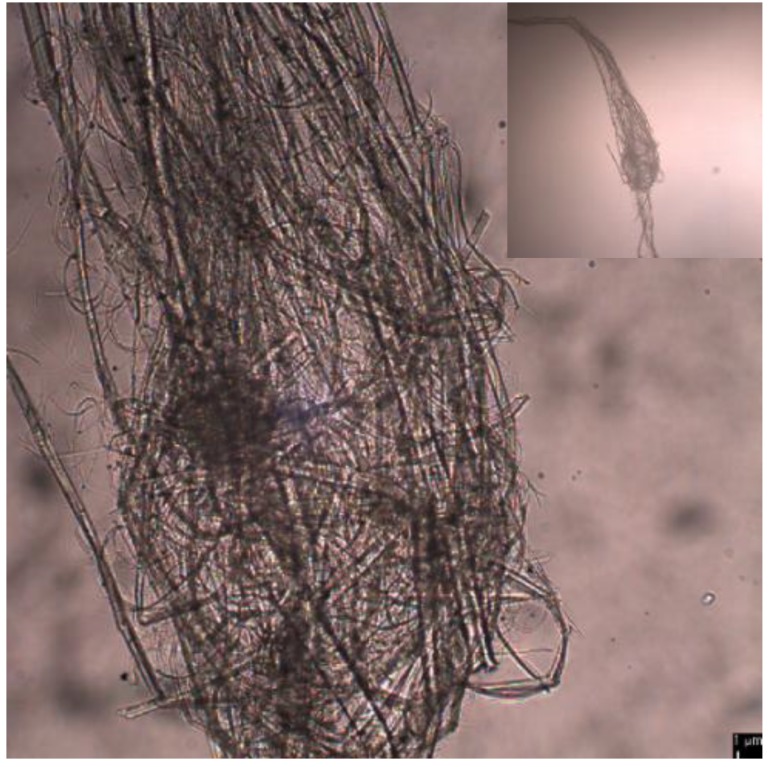
Optical microscopic image of entangled TEMPO-oxidized PMBFs after ultrasonic treatment at pH 3.

**Figure 5 polymers-11-00414-f005:**
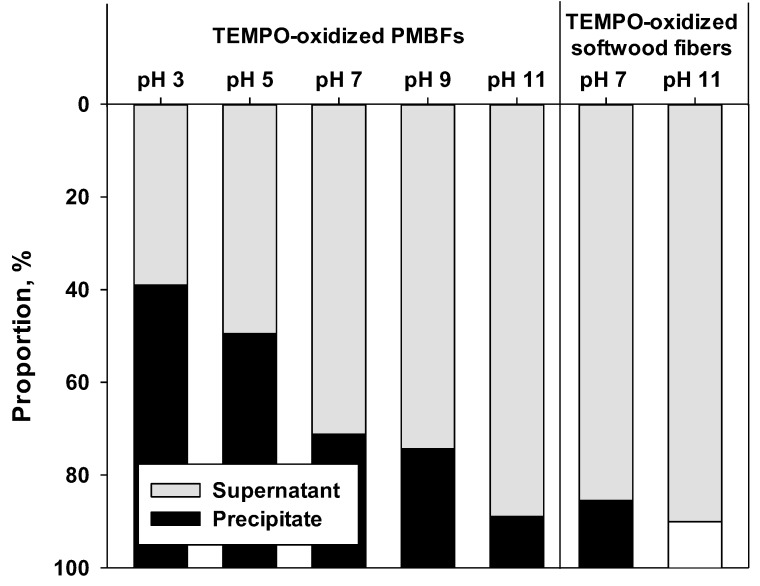
Proportion of supernatant and precipitate after centrifugation of the ultrasonicated suspension of TEMPO-oxidized PMBFs and softwood fibers.

**Figure 6 polymers-11-00414-f006:**
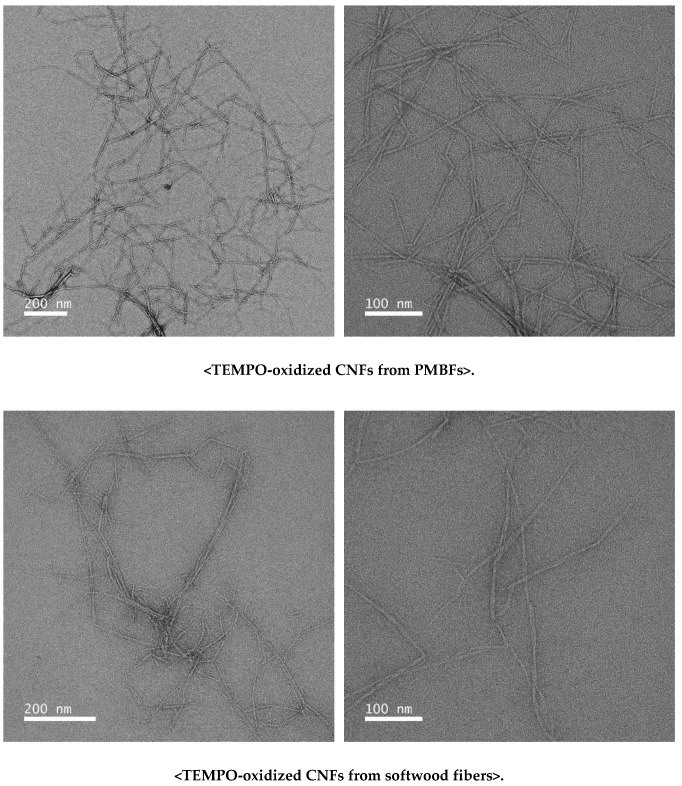
TEM images of TEMPO-oxidized CNFs from PMBFs and softwood fibers.

**Figure 7 polymers-11-00414-f007:**
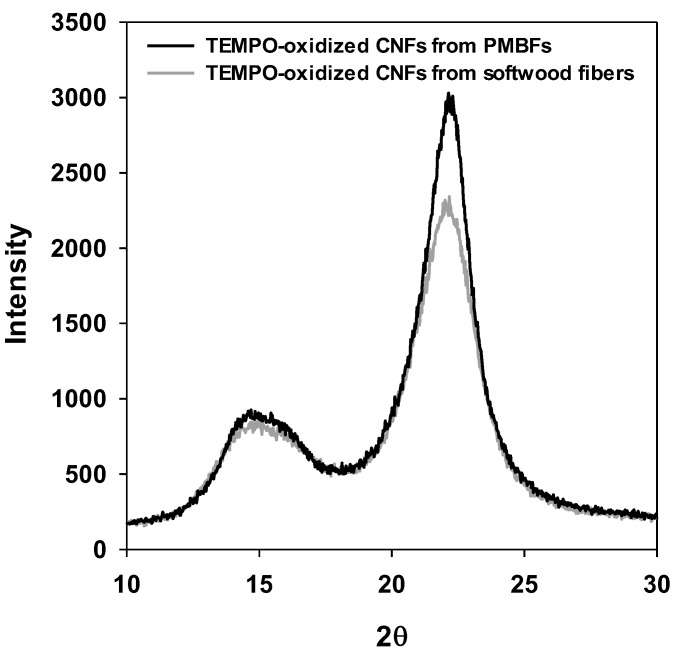
X-ray diffractograms of TEMPO-oxidized CNFs from PMBFs and softwood fibers.

**Table 1 polymers-11-00414-t001:** Tensile strength of nanopaper sheets formed by TEMPO-oxidized CNFs from PMBFs and softwood fibers.

Sample Form	Nanopaper Sheet	
Source of TEMPO-Oxidized CNFs	PMBFs	Softwood Fibers
Tensile strength (MPa)	110.18 ± 12.88	84.19 ± 10.69

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
