# Peer review of "Effects of pH on Nanofibrillation of TEMPO-Oxidized Paper Mulberry Bast Fibers"

_polymers, 2019, doi:10.3390/polym11030414_

Round 1
Reviewer 1 Report
1. Authors removed extractives after alkaline digestion using ethanol/benzene mixture. However, it is common to conduct the extraction before alkaline digestion because extractives consume quite amount of potassium hydroxide and then decrease the digestion efficiency. Please justify the reason why conducting extraction after digestion.
2. The concentration of TEMPO is only 1mmol/g cellulose which is quite low. Have authors tested the results at various TEMPO concentration? It is necessary to measure the carboxylate content of TEMPO-treated CNFs and evaluate the DP of CNFs after TEMPO treatment.
3. The mechanical properties of TEMPO-treated CNF sheets are lower than that in previous studies. There is a necessity to clarify the unsatisfied mechanical performance.
4. The cross-head speed (20mm /min) of universal tester is too fast for a 20mm span sample. Did specimens reach rupture point within 30 seconds to five minutes?
Author Response
Thank you for your critical reviews and valuable comments to improve our manuscript. We explained detail about your comments in attached Word file, and please kindly check.

Reviewer 2 Report
This paper describes the use of mulberry fibres to produce cellulose nanofibrils and an investigation of the effect of pH on nanofibrillation efficiency.
Overall, the paper is excellent. I would recommend the following changes:
- Specify number of samples used for tensile testing.
- Specify number of measurements taken to determine fibre lengths and diameters.
I would also like to praise the illustration of the dispersibility in Figure 3, which shows excellent presentation of results in a clear and understandable way, even for non-specialists.
Author Response

(The authors gave the same response as above.)

Round 2
Reviewer 1 Report
Authors can add one paragraph in materials and methods to describe the method to measure nanosheet density
Author Response
Thank you for your critical reviews and valuable comments to improve our manuscript. We revised the manuscript as your comments in attached Word file, and please kindly check.
